# Optimal Postoperative Surveillance Strategies for Colorectal Cancer: A Retrospective Observational Study

**DOI:** 10.3390/cancers13143502

**Published:** 2021-07-13

**Authors:** Min-Young Park, In-Ja Park, Hyo-Seon Ryu, Jay Jung, Min-Sung Kim, Seok-Byung Lim, Chang-Sik Yu, Jin-Cheon Kim

**Affiliations:** Department of Colon and Rectal Surgery, Asan Medical Center, University of Ulsan College of Medicine, Seoul 05505, Korea; pyong31@naver.com (M.-Y.P.); rang311@naver.com (H.-S.R.); jayandy15@naver.com (J.J.); bongkay4@gmail.com (M.-S.K.); sblim@amc.seoul.kr (S.-B.L.); csyu@amc.seoul.kr (C.-S.Y.); jckim@amc.seoul.kr (J.-C.K.)

**Keywords:** colorectal cancer, surveillance, recurrence, survival

## Abstract

**Simple Summary:**

Optimal surveillance strategies for colorectal cancer remain undetermined, with intensive surveillance not conferring significant survival benefits. This study aimed to assess whether surveillance intensity is associated with recurrence and survival in patients with colorectal cancer. This retrospective observational study showed that, although frequent postoperative surveillance did not improve overall survival or recurrence-free survival, surveillance improved post-recurrence survival. Analysis using a recurrence risk-prediction model showed that intensive surveillance improved both post-recurrence survival and overall survival in patients who were at high risk of recurrence. Thus, intensive surveillance does not improve overall survival and recurrence-free survival but can help improve post-recurrence survival by detecting early-stage recurrence or increasing the curative resection rate.

**Abstract:**

This study aimed to assess whether surveillance intensity is associated with recurrence and survival in colorectal cancer (CRC) patients. Overall, 3794 patients with pathologic stage I–III CRC who underwent radical surgery between January 2012 and December 2014 were examined. Surveillance comprised abdominopelvic computed tomography (CT) every 6 months and chest CT annually for 5 years. Patients who underwent more than and less than an average of three imaging examinations annually were assigned to the high-intensity (HI) and low-intensity (LI) groups, respectively. Demographics were similar in both groups. T and N stages were higher and perineural and lymphovascular invasion were more frequent in the HI group (*p* < 0.001 each). The mean overall survival (OS) was similar for both groups; however, recurrence-free survival (RFS) was longer (*p* < 0.001) and post-recurrence survival (PRS) was shorter (*p* = 0.024) in the LI group. In the multivariate analysis, surveillance intensity was associated with RFS (*p* < 0.001) in contrast to PRS (*p* = 0.731). In patients with high recurrence risk predicted using the nomogram, OS was longer in the HI group (*p* < 0.001). A higher imaging frequency in patients at high risk of recurrence could be expected to lead to a slight increase in PRS but does not improve OS. Therefore, rather than increasing the number of CT scans in high-risk patients, other imaging modalities or innovative approaches, such as liquid biopsy, are required.

## 1. Introduction

Periodic surveillance for tumor recurrence is recommended in patients who undergo resection for colorectal cancer (CRC), because early detection and treatment of recurrent lesions improve the likelihood of curative treatment and overall survival (OS) [1]. Moreover, surveillance facilitates the assessment of the quality of the primary surgery and detects metachronous tumors early.

CRC is the second most common cancer among Korean men, the fourth most common cancer among Korean women, and the third most common cause of cancer-related deaths in South Korea [2]. Despite these high rates of prevalence and mortality, CRC patients represent the second largest group of 5-year cancer survivors. The 5-year trend from 2013 to 2017 indicates that approximately 78% of CRC patients in Korea had resectable tumors with localized or regional disease. Similarly, CRC is the third most common cancer in the United States, with similar incidence and mortality rates; moreover, approximately 75% of these patients have localized or regional disease, making them candidates for curative surgery [3]. More than 90% of local recurrences appear within 5 years after surgery, with most appearing within 3 years [4,5]. Intensive surveillance during this period may improve the early detection of recurrence, as well as patient prognosis [6,7,8].

Optimal surveillance strategies have not yet been established, with systematic reviews and randomized trials providing inconclusive results regarding surveillance-related survival benefits [9,10,11]. Although intensive surveillance does not significantly increase survival rates [12,13,14], it increases the frequency of curative surgery for recurrent lesions [15,16,17,18]. Survival rates are higher in patients who undergo computed tomography (CT) and measurements of carcinoembryonic antigen (CEA) levels than in patients who do not undergo these investigations [9,15]. The lack of consistency among reports highlights the need to evaluate the survival benefits of intensive surveillance.

Thus, this study aimed to assess whether surveillance intensity was associated with the detection of recurrence and survival in CRC patients and investigate the effect of intensive surveillance on curative treatment outcomes in patients with recurrent CRC.

## 2. Results

### 2.1. Patient Characteristics

A review of medical records identified 3794 patients: 2450 (64.6%) in the low-intensity surveillance (LI) group, and 1344 (35.4%) in the high-intensity surveillance (HI) group (Appendix A). Patient demographic characteristics and tumor clinicopathological features are shown in Table 1. Demographic characteristics did not differ between the LI and HI groups. T and N stages were higher in the HI than in the LI group, with the HI group having more risk factors, including higher degrees of malignant differentiation, perineural invasion (PNI), and positive circumferential resection margin (CRM). The average number of CT examinations per year was approximately 2-fold higher in the HI group than in the LI group (*p* < 0.001).

### 2.2. Oncologic Outcomes According to Surveillance Intensity

The average numbers of abdominopelvic CT (APCT; 2.35 ± 1.19 vs. 1.81 ± 3.31, *p* < 0.001) and chest CT (CCT; 1.38 ± 1.09 vs. 1.13 ± 2.45, *p* < 0.001) examinations were significantly higher in patients who experienced tumor recurrence than in those who did not. Patients who experienced recurrence were divided into those with intra-abdominal and intra-thoracic recurrence. The number of APCT examinations was higher in patients who experienced intra-abdominal recurrence (2.39 ± 1.09 vs. 1.83 ± 3.25, *p* < 0.001), whereas the number of CCT examinations was higher in patients who experienced intra-thoracic recurrence (1.71 ± 3.36 vs. 1.13 ± 2.37, *p* < 0.001). Among patients with rectal cancer, 27 had local recurrence, including 11 (40.7%) and 16 (59.3%) in the LI and HI groups, respectively (*p* = 0.074). However, the incidence of local recurrence differed significantly according to APCT intensity (*p* = 0.008), with 13 (1.2%) and 14 (3.3%) patients in the low and high APCT intensity groups, respectively, having local recurrence. Distant metastasis was confirmed in 444 patients: 180 (7.3%) in the LI group and 264 (19.6%) in the HI group (*p* < 0.001).

Recurrence-free interval (RFS) was significantly longer in the LI group than in the HI group (53.46 ± 23.07 vs. 49.88 ± 24.84 months, *p* < 0.001). The mean RFS did not differ between patients in the LI and HI groups who experienced recurrence (20.39 ± 14.87 vs. 20.19 ± 14.26 months, *p* = 0.888). Intra-abdominal RFS was longer in patients with low APCT intensity than in those with high APCT intensity (22.13 ± 15.39 vs. 18.65 ± 13.90 months, *p* = 0.047), whereas intra-thoracic RFS did not differ in patients with low and high CCT intensity (24.12 ± 14.59 vs. 19.62 ± 14.63 months, *p* = 0.067) (Figure 1A).

Tumors recurred in 180 and 264 patients in the LI and HI groups, respectively. Among patients who experienced recurrence, the mean post-recurrence survival (PRS) was significantly shorter in the LI group than in the HI group (29.77 ± 23.89 vs. 34.89 ± 22.79 months, *p* = 0.024). PRS, however, did not differ in patients with colon and rectal tumors (32.48 ± 25.45 vs. 33.15 ± 21.05 months, *p* = 0.762). Curative treatment was possible in 223 patients: 94 (52.2%) in the LI group and 129 (48.7%) in the HI group (*p = 0*.499). Among the 94 patients in the LI group, 47 (50.0%) each had colon and rectal cancers. Among the 129 patients in the HI group, 60 (46.5%) and 69 (53.5%) had colon and rectal cancers, respectively. Curative resection rates did not differ significantly between patients in the LI and HI groups with colon (*p* = 0.591) and rectal (*p* = 0.581) cancer. PRS was significantly longer in patients who did than in those who did not undergo curative resection after recurrence (43.04 ± 23.27 vs. 24.51 ± 19.91 months, *p* < 0.001) (Figure 1B).

The mean OS did not differ between the LI and HI groups (56.77 ± 21.17 vs. 57.59 ± 21.44 months, *p = 0*.259), with OS being similar in LI and HI patients with colon (55.41 ± 22.60 vs. 56.67 ± 23.19 months, *p = 0*.229) and rectal (59.04 ± 18.33 vs. 58.59 ± 19.31 months, *p* = 0.453) cancers (Figure 1C).

When subgroup analysis was performed according to the stage, the higher the stage, the higher the percentage of patients in the HI group. In addition, the recurrence rate was high in the HI group regardless of stage, but such a high recurrence rate tended not to affect the survival improvement (Appendix A).

### 2.3. Factors Associated with Oncologic Outcomes

Multivariate analyses revealed that age, sex, surveillance intensity, differentiation, pathologic T and N stage, PNI, and CRM were significantly associated with OS and that age, surveillance intensity, differentiation, pathologic T and N stage, PNI, and CRM were significantly associated with RFS (Table 2 and Appendix A).

Univariate analyses showed that age, differentiation, pathologic N stage, lymphovascular invasion (LVI), and curative resection were significantly associated with PRS in patients who experienced tumor recurrence, with the multivariate analysis showing that age, differentiation, pathologic N stage, and curative resection were significantly associated with PRS (Table 2). Separate analyses of factors associated with intra-abdominal and intra-thoracic recurrence showed that curative resection was significantly associated with PRS in both subpopulations (Appendix A).

### 2.4. Oncologic Outcomes According to the Prediction Model

Results of the validation and univariate and multivariate regression analysis models are shown in Appendix A. Both univariate and multivariable analyses showed that sex, (y)pT stage, (y)pN stage, and PNI were significant predictors of recurrence. These four variables were used to construct a nomogram for predicting recurrence (Figure 2). To use the nomogram, a vertical line was drawn to the top row and points were assigned for each variable. The total number of points was calculated, and a vertical line was drawn downward from the top row to determine the probability of recurrence. Predictive accuracy measured using the c-index was 0.74. The nomogram was well calibrated, with good correlation and no deviation between the predicted and validated outcomes across the spectrum of predictions. In the external validation group (*n* = 2215), the accuracy of the model was 0.73. Calibration plots of nomogram-predicted probabilities and actual recurrence probability in the external validation group are shown in Appendix A.

According to the prediction model, oncologic outcomes were compared by dividing the study population into the high-recurrence risk (HR) and low-recurrence risk (LR) groups. RFS was longer in the LI group with LR patients (55.66 ± 21.15 vs. 53.08 ± 23.76 months, *p* = 0.009), although the surveillance intensity did not affect the RFS in HR patients (45.48 ± 27.57 vs. 46.91 ± 25.45 months, *p* = 0.352) (Figure 3A). PRS was shorter in the LI group regardless of the recurrence risk, although the *p*-values were marginally significant (LR, 34.04 ± 25.12 vs. 41.62 ± 21.95, *p* = 0.051; HR, 26.50 ± 22.48 vs. 32.42 ± 22.64, *p* = 0.033). Furthermore, the 2-year RFS rate was lower in the HI group regardless of the recurrence risk (LR, 58.9% vs. 77.4%, *p* = 0.022; HR, 45.1% vs. 59.1%, *p* = 0.027) (Figure 3B). There was no difference in OS according to the surveillance intensity in LR patients (58.20 ± 19.53 vs. 58.86 ± 20.84 months, *p* = 0.477), but OS was significantly shorter in the HI group with HR patients (OS: 51.58 ± 25.64 vs. 56.40 ± 21.92 months, *p* < 0.001) (Figure 3C).

## 3. Discussion

This study showed that intensive postoperative surveillance could enhance PRS duration and PRS rates. In contrast, postoperative surveillance did not enhance RFS or OS, with the exception that it improved OS in patients classified as high risk of recurrence in the recurrence risk-prediction model.

The guidelines for CRC patients recommend surveillance after primary surgery with a curative intent [19,20,21,22,23], although these guidelines differ in their recommendations. For example, the European Society of Medical Oncology recommends an abdominal and chest CT every 6–12 months for 3 years, and then yearly for 2 years in colon cancer patients but has no imaging recommendations for rectal cancer patients. The American Society of Clinical Oncology recommends abdominal and chest CT every 6–12 months for the first 3 years for high-risk patients and every 12 months for the first 3 years in all other patients. The National Comprehensive Cancer Network recommends abdominal CT for high-risk patients with poorly differentiated cancer or those with perineural or venous invasion, although there are no guidelines with regard to its frequency. The American Society of Colorectal Surgeons guidelines recommend chest and abdominopelvic imaging annually for 5 years.

Intensive surveillance has shown associations with significant reductions in mortality [24,25,26,27,28]. However, although an intensive surveillance program after curative treatment for CRC detected asymptomatic local or distant recurrences, it did not affect OS [29]. Similarly, a randomized trial changed the original endpoint of unmeasured OS to a practical endpoint of surgical treatment for recurrence with curative intent [16]. Several meta-analyses and prospective randomized trials have failed to show that intensive surveillance is associated with survival benefits [15,18,30].

In this study, patients were divided into the LI and HI groups according to the number of imaging examinations during follow-up. The average number of imaging examinations was higher in patients with recurrence than in those without recurrence, regardless of its location. Patients in the HI group had higher pathologic T and N stages and were more likely to have risk factors, including LVI and PNI, suggesting that surveillance was more frequent in patients with high risks of recurrence. These factors could cause a selection bias, which may have influenced the study outcome. However, this was corrected using a recurrence risk-prediction model. Among rectal cancer patients, 47 patients (18 in the LI group and 29 in the HI group) had local recurrences, mostly to the anastomotic site and pelvic lymph nodes, with the rate of local recurrence being significantly higher in the HI group than in the LI group (2.2% vs. 0.7%, *p* < 0.001). Among the 47 patients, 37 underwent surgical resection, with 31 (12 (80%) in the LI group and 19 (86%) in the HI group) achieving curative resection (*p* = 0.827). These similar rates may be because of the small number of patients with local recurrence. Among the remaining 10 patients, one refused treatment, whereas the other nine received palliative treatments, including chemotherapy or radiotherapy, owing to combined systemic metastasis or invasion of other organs.

RFS was longer in the LI group than in the HI group, whereas PRS was longer in the HI group than in the LI group, with the OS being similar in the two groups. Shorter RFS may be due to the more aggressive behavior of tumors in the HI group. The surveillance intensity was not significantly associated with survival in patients with either colon or rectal cancer. Pathologic risk factors, including the histologic degree of differentiation, pathologic T and N stage, and PNI, had greater effects on OS and RFS as reported [31]. Moreover, differentiation and pathologic N stage affected PRS. In particular, curative resection had a greater effect on PRS than surveillance intensity. The PRS of patients with recurrence was longer in those who underwent curative resection than in those who did not (43 vs. 24 months, *p* < 0.001), with the multivariate analysis confirming that curative resection improved PRS.

A model predicting recurrence according to risk factors was developed to correct for selection bias owing to the aggressive biology of the HI group, with patients in the LI and HI groups each further assigned to subgroups with low and high risks of recurrence. The RFS of patients with LR was shorter in the HI group than in the LI group, whereas the PRS was longer in the HI group than in the LI group. Consequently, the OS in the HI and LI groups was similar. The RFS in patients with HR was similar in the LI and HI groups, whereas PRS was longer in the HI group than in the LI group. Analysis showed that a higher percentage of patients in the HI group (57.8%) than in the LI group underwent curative resection at the time of recurrence detection, with 64.7% of patients with HR being in the HI group. These findings suggested that intensive surveillance of patients with HR after surgery could result in a similar curative resection rate to that of patients with LR. However, the OS in the HI group with HR patients was shorter than that in the LI group. When only patients with HR were analyzed, the overall patient characteristics were similar between the LI and HI groups, but the recurrence rate was high in the HI group (Appendix A). Thus, we infer that other clinical factors, which caused surgeons to perform intensive surveillance, influenced the recurrence rate and OS in addition to the pathologic risk factors included in calculating the recurrence risk in the nomogram.

This study had some limitations. First, this was a retrospective observational cohort study, with patients not being randomized. Surveillance intensity can vary according to patients’ condition at the time of treatment, which may have resulted in selection bias. To analyze the impact of surveillance intensity according to risk group, we tried to categorize patients into risk group considering pathologic risk factors, but, clinical factors which may have an influence on surveillance intensity were not sufficiently considered in the nomogram. Therefore, the high intensity group may include patients who need caution for recurrence, regardless of risk group. It may act as confounding factors for analysis. Second, the surveillance schedules could be physician dependent. Additional research is needed to determine the standard routine surveillance in our institution.

## 4. Materials and Methods

### 4.1. Patients and Clinical Variables

Adults who underwent radical resection and elective surgery for primary CRC and those who received preoperative chemoradiotherapy (PCRT) followed by radical resection from January 2012 to December 2014 at Asan Medical Center, Seoul, Korea were retrospectively evaluated. All patients who underwent radical surgery for primary CRC and were identified as having pathologic stage I to III CRC after surgery were enrolled in this study. Children and adolescents younger than 18 years were not included in this study because they underwent surgery by a pediatric surgeon. Patients with synchronous distant metastases, synchronous cancer in another organ, cancer diagnosed within 5 years, or inflammatory disease-associated CRC; those who underwent local excision; those with unknown staging status; or those who were lost to follow-up were excluded. The patient characteristics analyzed included age, sex, pathologic differentiation, LVI, PNI, CRM of rectal cancer (involving < 1 mm), PCRT, recurrence, treatment after recurrence, and survival. Postoperative surveillance included APCT and CCT.

This study was approved by the Institutional Review Board of Asan Medical Center (registration no. 2017-0955), which waived the requirement for informed consent because of the retrospective observational nature of this study that involved the analysis of previously collected data.

### 4.2. Treatment and Surveillance

The objectives of surgical treatment for colon cancer were ligation of the feeding vessels at their roots, removal of the principal nodes, and achievement of sufficient proximal and distal resection margins. The objective of surgery for rectal cancer was total mesorectal excision. Patients who received PCRT underwent surgical resection 6–10 weeks after completing PCRT. Most surgical procedures were performed by one of seven experienced colorectal surgeons, with the remaining procedures performed by colorectal fellows.

Adjuvant chemotherapy has been recommended for pathologic stage III colon cancer patients and stage II colon cancer patients with risk factors, including preoperative obstruction, LVI, PNI, high tumor budding, and <12 resected lymph nodes. Adjuvant chemotherapy was recommended for pathologic stage II and III rectal cancer patients and those treated with PCRT, regardless of the pathologic stage. PCRT was indicated for clinical stage II or III cancer patients, clinical stage I cancer patients eligible for sphincter-saving surgery owing to low-lying rectal cancer, and patients who were not candidates for major surgeries because of medical comorbidities.

All patients underwent physical examination, and serum CEA levels were measured every 3–6 months postoperatively. Abdominal, pelvic, and chest CTs were performed every 6–12 months. Patients with obstructive lesions underwent colonoscopy within 6 months after surgical resection and, thereafter, every 2–3 years.

### 4.3. Definition of Surveillance Intensity

All patients were followed up with APCT and CCT for approximately 5 years after surgery. Patients underwent surveillance every 6 months at the outpatient clinic, including APCT every 6 months and CCT every 12 months on an average. The number of expected imaging examinations per year was three: two APCT and one CCT. Imaging examinations were performed more frequently in patients with high pathologic stage or high-risk factors and in patients with findings on previous examinations that were ambiguous for metastasis.

The average number of imaging examinations per patient was calculated as the number of examinations during the 5-year follow-up in patients without recurrence or as the number of examinations until the first recurrence for patients experiencing recurrence. Patients who underwent either more than and less than three examinations per year were classified into the HI and LI groups, respectively.

### 4.4. Statistical Analyses

Continuous variables are expressed as the mean ± standard deviation (SD) and were compared using Student’s *t*-tests. Categorical variables are expressed as numbers and percentages and were compared using Pearson’s chi-square test or Fisher’s exact test. Univariate analyses were performed to identify factors associated with survival. Factors with *p* < 0.1 in the univariate analysis were included in the multivariate binary logistic regression analysis. The end points of interest were OS, RFS, and PRS. OS was defined as the period from the date of surgery to the date of death. RFS was defined as the period from the date of surgery to the date of first recurrence. PRS in patients with recurrence was defined as the period from first recurrence to death. Furthermore, each death and recurrence was classified as an event. OS, RFS, and PRS were calculated using the Kaplan–Meier method and compared using the Cox regression test. All statistical analyses were performed using SPSS for Windows, ver. 25.0 (SPSS Inc., Chicago, IL, USA), with *p* < 0.05 defined as indicating statistical significance.

To develop a model predictive of RFS, a Cox proportional hazard regression model was fitted to the data from the development cohort. The potential predictors of RFS that were analyzed included age, sex, differentiation, LVI, PNI, T stage, and N stage. Variables for the final prediction model were selected using the bootstrapping resampling method (500 repetitions), which determines the predictive robustness of candidate variables. A 50% relative frequency of selection and clinical relevance were the criteria for inclusion of the variable in the final analysis model.

The final models were validated according to their discrimination and calibration. Discrimination was assessed using Harrell’s C-index and time-dependent area under the receiver operating characteristic curves to predict 1-, 3-, and 5-year recurrences. Calibration was assessed using calibration plots at 1, 3, and 5 years. Both discrimination and calibration were evaluated in the development and validation cohorts.

The final models are presented as nomograms, which provide prognostic indices and corresponding 1-, 3-, and 5-year RFS probabilities. The formulas for 1-, 3-, and 5-year RFS probabilities of the models are presented as lookup tables for future studies that undertake fully independent model validation.

The R Package (version 3.6.0) was used for statistical analyses of the prediction model and nomograms.

## 5. Conclusions

Recurrence was detected more commonly in patients with colorectal cancer who received frequent postoperative imaging surveillance, but RFS and overall survival was not associated with surveillance intensity in the overall cohort. In the high risk group, however, frequent imaging surveillance was related with improved RPS although it was not translated into improvement of OS. Therefore, in high-risk patients, intensive surveillance with CT might be carefully evaluated in terms of oncologic benefit balancing with medical economics. There is a need to explore other imaging modalities or innovative approaches, such as liquid biopsy, and evaluate the effect of these tools on oncologic outcomes and facilitate the early detection of recurrence.

## Figures and Tables

**Figure 1 cancers-13-03502-f001:**
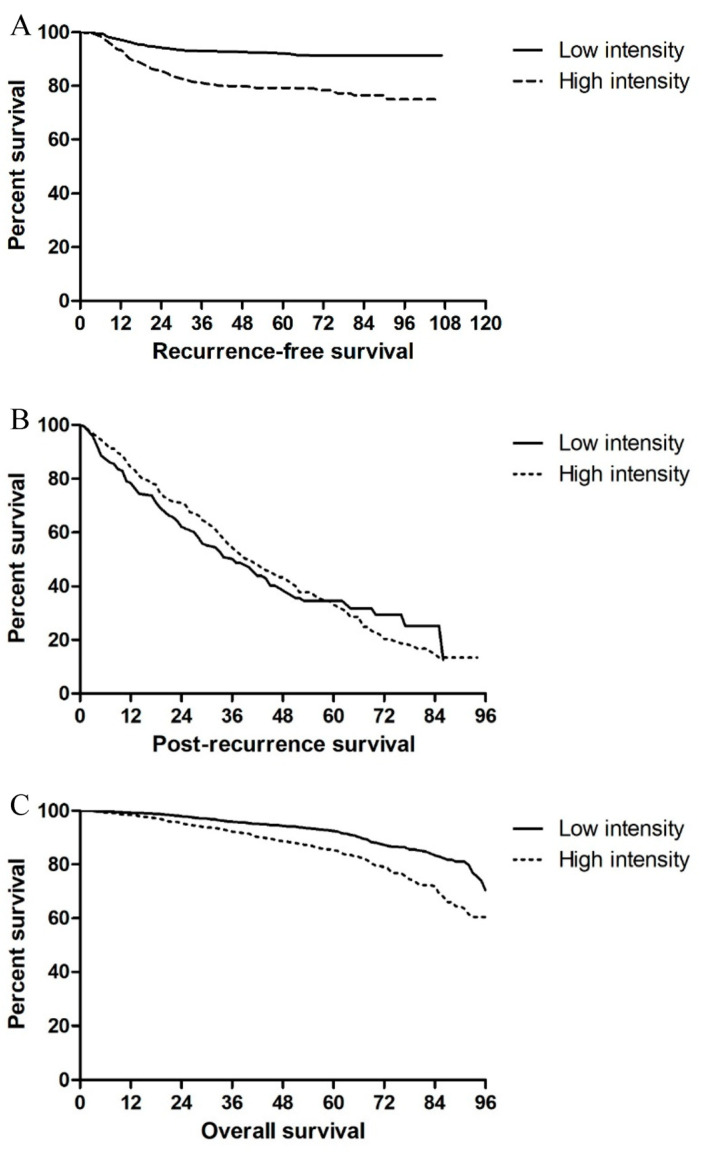
Kaplan–Meier analyses of (**A**) recurrence-free survival, (**B**) post-recurrence survival, and (**C**) overall survival according to surveillance intensity. Abbreviations: LI, low surveillance intensity; HI, high surveillance intensity; LR, low-recurrence risk; HR, high-recurrence risk.

**Figure 2 cancers-13-03502-f002:**
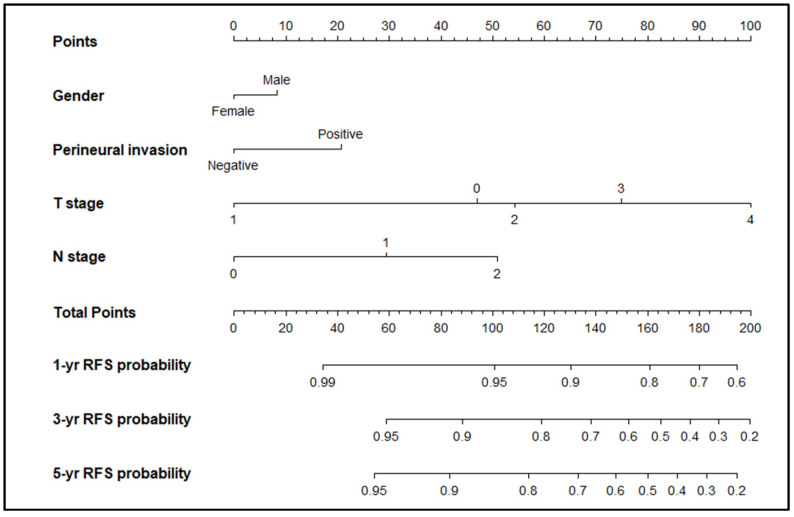
Nomogram predicting the probability of recurrence after radical surgery for colorectal cancer.

**Figure 3 cancers-13-03502-f003:**
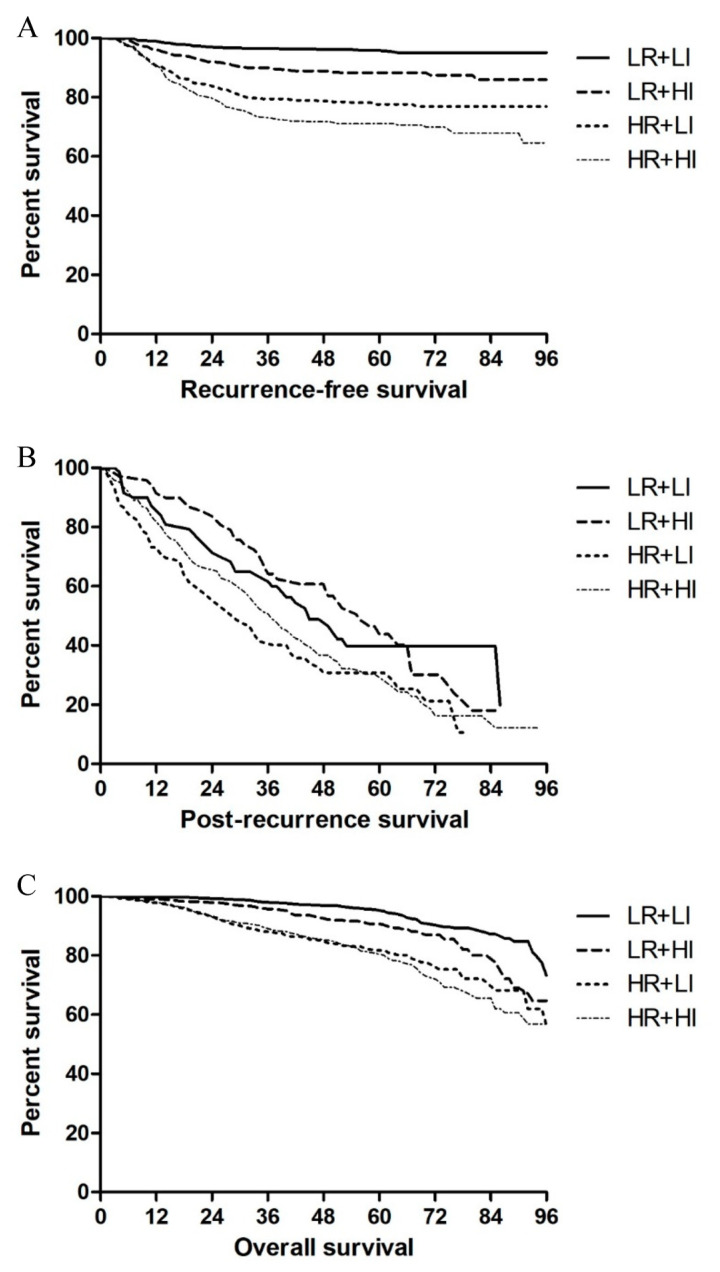
Kaplan–Meier analyses of (**A**) recurrence-free survival, (**B**) post-recurrence survival, and (**C**) overall survival according to surveillance intensity and predicted recurrence risk. Abbreviations: LI, low surveillance intensity; HI, high surveillance intensity; LR, low-recurrence risk; HR, high-recurrence risk.

**Table 1 cancers-13-03502-t001:** Demographic and clinical characteristics of participants (*N* = 3794).

Variables	Surveillance Intensity		*p*-Value
	Lower Intensity (*N* = 2450)	Higher Intensity (*N* = 1344)	
Age, mean (IQR) (years)	61.9 (54.0–71.0)	60.1 (53.0–69.0)	0.964
Sex, *N* (%)			0.063
Male	1407 (57.4)	814 (60.6)	
Female	1043 (42.6)	530 (39.4)	
Cancer site, *N* (%)			<0.001
Colon	1535 (62.7)	703 (52.3)	
Rectum	915 (37.3)	641 (47.7)	
Differentiation, *N* (%)			<0.001
WD/MD	2325 (94.9)	1218 (90.6)	
PD/SRC/MUC	125 (5.1)	126 (9.4)	
Total lymph nodes, *N* (%)			0.500
<12	97 (4.0)	61 (4.5)	
≥12	2353 (96.0)	1283 (95.5)	
(y)pT, *N* (%)			<0.001
0	146 (6.0)	56 (4.2)	
1	600 (24.5)	89 (6.6)	
2	469 (19.1)	181 (13.5)	
3	1119 (45.7)	852 (63.3)	
4	116 (4.7)	166 (12.4)	
(y)pN, *N (*%)			<0.001
0	1917 (78.2)	681 (50.7)	
1	434 (17.7)	496 (36.9)	
2	99 (4.0)	167 (12.4)	
Perineural invasion, *N* (%)	265 (10.8)	304 (22.6)	<0.001
Lymphovascular invasion, *N* (%)	537 (21.9)	479 (35.6)	<0.001
Resection margin, *N* (%)	29 (1.2)	38 (2.8)	<0.001
APCT, mean (SD)	1.41 (0.49)	2.73 (5.12)	<0.001
CCT, mean (SD)	0.82 (0.41)	1.77 (3.80)	<0.001
Total imaging studies, mean (SD)	2.23 (0.71)	4.50 (8.77)	<0.001

IQR, interquartile range; *N*, number; WD, well differentiated; MD, moderately differentiated; PD, poorly differentiated; SRC, signet ring cell type; MUC, mucinous carcinoma; APCT, abdominopelvic computed tomography; SD, standard deviation; CCT, chest computed tomography.

**Table 2 cancers-13-03502-t002:** Factors affecting survival of participants.

Factors	Univariate		Multivariate	
	HR (95% CI)	*p*	HR (95% CI)	*p*-Value
Overall Survival				
Age (years)	1.059 (1.049–1.069)	<0.001	1.061 (1.051–1.070)	<0.001
Sex	0.779 (0.646–0.940)	0.009	0.766 (0.635–0.925)	0.006
Surveillance intensity	1.849 (1.545–2.214)	<0.001	1.454 (1.201–1.760)	<0.001
Differentiation				
WD/MD	Ref		Ref	
PD/SRC/MUC	2.479 (1.890–3.252)	<0.001	1.983 (1.504–2.616)	<0.001
(y)pT stage				
0–2	Ref		Ref	
3–4	2.743 (2.198–3.423)	<0.001	1.718 (1.345–2.195)	<0.001
(y)pN stage				
0	Ref		Ref	
1	1.924 (1.571–2.356)	<0.001	1.482 (1.191–1.845)	<0.001
2	3.760 (2.935–4.817)	<0.001	2.344 (1.779–3.089)	<0.001
Lymphovascular invasion	1.885 (1.567–2.268)	<0.001	1.118 (0.911–1.372)	0.284
Perineural invasion	1.919 (1.560–2.361)	<0.001	1.324 (1.061–1.653)	0.013
Resection margin	3.016 (1.965–4.629)	<0.001	2.140 (1.387–3.301)	0.001
Recurrence-free survival				
Age (years)	1.006 (0.998–1.014)	0.165	1.010 (1.002–1.019)	0.014
Sex	0.884 (0.730–1.071)	0.207		
Surveillance intensity	2.801 (2.317–3.385)	<0.001	1.700 (1.391–2.077)	<0.001
Differentiation				
WD/MD	Ref		Ref	
PD/SRC/MUC	1.848 (1.373–2.487)	<0.001	1.361 (1.007–1.839)	0.045
(y)pT stage				
0–2	Ref		Ref	
3–4	4.491 (3.452–5.843)	<0.001	2.351 (1.766–3.130)	<0.001
(y)pN stage				
0	Ref		Ref	
1	3.012 (2.442–3.714)	<0.001	1.825 (1.454–2.290)	<0.001
2	6.217 (4.839–7.988)	<0.001	2.906 (2.190–3.858)	<0.001
Lymphovascular invasion	2.212 (1.833–2.669)	<0.001	1.045 (0.847–1.289)	0.684
Perineural invasion	3.235 (2.658–3.938)	<0.001	1.743 (1.408–2.159)	<0.001
Resection margin	4.402 (2.982–6.498)	<0.001	2.471 (1.667–3.664)	<0.001
Post-recurrence survival				
Age (years)	1.022 (1.011–1.033)	<0.001	1.026 (1.015–1.037)	<0.001
Sex	1.048 (0.818–1.343)	0.709		
Image intensity	0.957 (0.745–1.229)	0.731		
Differentiation				
WD/MD	Ref		Ref	
PD/SRC/MUC	3.724 (2.644–5.246)	<0.001	2.493 (1.742–3.568)	<0.001
(y)pT stage				
0–2	Ref			
3–4	1.220 (0.854–1.743)	0.274		
(y)pN stage				
0	Ref		Ref	
1	1.297 (0.979–1.719)	0.070	1.385 (1.030–1.864)	0.031
2	1.803 (1.316–2.469)	<0.001	1.607 (1.120–2.305)	0.010
Lymphovascular invasion	1.516 (1.188–1.935)	0.001	1.077 (0.821–1.412)	0.592
Perineural invasion	1.032 (0.799–1.344)	0.809		
Resection margin	1.340 (0.840–2.138)	0.220		
Curative resection	0.197 (0.148–0.264)	<0.001	0.218 (0.162–0.292)	<0.001

HR, hazard ratio; CI, confidence interval; WD, well differentiated; MD, moderately differentiated; PD, poorly differentiated; SRC, signet ring cell type; MUC, mucinous carcinoma; Ref, reference.

## Data Availability

The data presented in this study are available on request from the corresponding author. The data are not publicly available due to conditions of the ethics committee of our university.

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
