# Peer review of "Optimal Postoperative Surveillance Strategies for Colorectal Cancer: A Retrospective Observational Study"

_cancers, 2021, doi:10.3390/cancers13143502_

Round 1
Reviewer 1 Report
In this manuscript by Park M et al, the author assessed the correlation between the surveillance intensity and survival of CRC patients. This manuscript includes almost 4000 patients for the analysis and the number seems to be large. Furthermore, the author created the nomogram according their findings and validated with the external data set which seems to be valid method. There are some interesting findings in the manuscript, however there are several concerns. Below are my comments.
- Overall, the manuscript includes sufficient number of cases to assess the main theme. My biggest question and concern is why does the high intensity (HI) surveillance associates with poor survival outcome? In Figure 2, I see that the HI group would have shorter RFS and OS than Low intensity (LI) group, but I agree to the author that HI group includes more patients with risk factors and the selection bias seems to affect the result. However, looking into Table 2 and 3, survival intensity is shown to be significant risk factor from the multivariate analysis. I do not see the reference so that I assume that the reference is LI. It would make sense to me if the HR would be lower than 1, which means that HI would be better than LI, but I don't understand why HI would lead to worse survival regardless of the TNM factor and other risk factors. Does the frequent CT scan would create secondary cancer? This point should be explained clearly.
- Another critical point would be in Figure 5. It is very interesting that the author used external data set for the validation of nomogram and compared the low and high risk patient with HI and LI. However, why does the survival differ between the LI and HI in same low risk group and high risk group? It would make sense if the LI and HI would be totally same in low risk group, or HI group would be better than LI group in high risk patients, for example. Presented data suggests that the HI surveillance has a significant risk to aggravate the tumor progression, however I do not have any explanation why the surveillance, not the pathological or clinical factors, would affect the survival.
- I would recommend the author to select the key data and remove some of the Table or move them to supplement since the data appears too many and hard track them all.
- In Figure 1A, the symbol of LI and HI seems to be opposite compared to Figure 1B and 1C. Please correct it.
- I would suggest the author to give some idea what kind of surveillance would be more feasible according to the present study.
Author Response
Response to Reviewer 1 Comments
We appreciate your constructive comments and the valuable questions that have helped us to improve our manuscript. In our point-by-point responses that follow, we have described the changes that we have made to the original manuscript in view of your comments and suggestions.
Point 1: Overall, the manuscript includes sufficient number of cases to assess the main theme. My biggest question and concern is why does the high intensity (HI) surveillance associates with poor survival outcome? In Figure 2, I see that the HI group would have shorter RFS and OS than Low intensity (LI) group, but I agree to the author that HI group includes more patients with risk factors and the selection bias seems to affect the result. However, looking into Table 2 and 3, survival intensity is shown to be significant risk factor from the multivariate analysis. I do not see the reference so that I assume that the reference is LI. It would make sense to me if the HR would be lower than 1, which means that HI would be better than LI, but I don't understand why HI would lead to worse survival regardless of the TNM factor and other risk factors. Does the frequent CT scan would create secondary cancer? This point should be explained clearly.
Response 1: Thank you for your very insightful comments. We completely understand your concern and the need for clarity. This is exactly what we intend to understand and solve. In our opinion, the implication of the association in the sentence “high intensity (HI) surveillance is associated with poor survival outcome” is not the “influence” but rather a “phenomenon.” As you indicated, the HI group includes more patients with pathologic risk factors. In addition to the risk factors included in the multivariate analysis, the clinical risk factors that are generally based on surgical findings, such as surgical T stage, bowel obstruction, and inflammatory adhesion to adjacent organs, affected the surveillance intensity. These clinical risk factors might not have been sufficiently adjusted in the analysis. Moreover, we considered the factor of surveillance intensity to be a result of the abovementioned clinical factors rather than an effect of the number of CT scans. Nonetheless, this does not mean that HI influenced recurrence but instead implies that the group with high-intensity follow-up was prone to recurrence.
These results were considered to be in the context of your question #2, which we have addressed in the revised manuscript in the description of the subgroup analysis.
Point 2: Another critical point would be in Figure 5. It is very interesting that the author used external data set for the validation of nomogram and compared the low and high risk patient with HI and LI. However, why does the survival differ between the LI and HI in same low risk group and high risk group? It would make sense if the LI and HI would be totally same in low risk group, or HI group would be better than LI group in high risk patients, for example. Presented data suggests that the HI surveillance has a significant risk to aggravate the tumor progression, however I do not have any explanation why the surveillance, not the pathological or clinical factors, would affect the survival.
Response 2: Thank you for specifying this concern. As mentioned in our response to Question 1, the HI group comprises patients whom the surgeons suspected would have both clinical and pathologic risk factors. Although we categorized the risk groups based on pathologic risk factors, we could not present this point very clearly. When patients in the same recurrence risk group underwent subgroup analysis according to surveillance intensity, the recurrence rate was higher in the HI group, despite no difference in the TNM stage and other pathologic risk factors, especially in the HR group.
Table S5 Demographic and clinical characteristics of patients with high recurrence risk (N=1228).
|
Variables |
Surveillance intensity |
P-value |
|
|
Lower intensity (n=530) |
Higher intensity (n=698) |
||
|
Differentiation, No (%) |
|
|
.047 |
|
WD/MD |
486 (91.7) |
615 (88.1) |
|
|
PD/SRC/MUC |
44 (8.3) |
83 (11.9) |
|
|
Total lymph nodes, No (%) |
|
|
.840 |
|
<12 |
10 (1.9) |
15 (2.1) |
|
|
≥12 |
520 (98.1) |
683 (97.9) |
|
|
(y)pT, No (%) |
|
|
.736 |
|
2 |
7 (1.3) |
9 (1.3) |
|
|
3 |
407 (76.8) |
523 (74.9) |
|
|
4 |
116 (21.9) |
166 (23.8) |
|
|
(y)pN, No (%) |
|
|
.052 |
|
0 |
108 (20.4) |
119 (17.0) |
|
|
1 |
324 (61.1) |
413 (59.2) |
|
|
2 |
98 (18.5) |
166 (23.8) |
|
|
Perineural invasion, No (%) |
194 (36.6) |
268 (38.4) |
.552 |
|
Lymphovascular invasion, No (%) |
263 (49.6) |
357 (51.1) |
.604 |
|
Resection margin, No (%) |
17 (3.2) |
28 (4.0) |
.540 |
|
Total imaging studies, mean (SD) |
2.31 (0.85) |
4.44 (7.27) |
<.001 |
|
Overall recurrence, No (%) |
102 (19.2) |
193 (27.7) |
.001 |
|
Local recurrence, No (%) |
7 (1.3) |
20 (2.9) |
.078 |
|
Systemic recurrence, No (%) |
95 (17.9) |
173 (24.8) |
.004 |
Therefore, this result may indicate that not only the TNM stage and pathologic risk factors but also the clinical operation findings, such as obstruction as well as difficult and incomplete TME that necessitated intensive surveillance, may have influenced the surveillance intensity. We hope that this response and the response to Question #1 clarify this point adequately. The issue raised in your comment is a very critical aspect of this study, and we have attempted to further elucidate it in the Discussion section and supplementary materials (Page 9, line 7 & Table S5).
Point 3: I would recommend the author to select the key data and remove some of the Table or move them to supplement since the data appears too many and hard track them all.
Response 3: Thank you for your valuable suggestion. Accordingly, we have organized the key data into two tables and three figures, and the remaining data were moved to the supplementary materials. The change in the number of tables, figures, and supplementary materials has been indicated in the manuscript.
Point 4: In Figure 1A, the symbol of LI and HI seems to be opposite compared to Figure 1B and 1C. Please correct it.
Response 4: Thank you for your comment. Accordingly, we have corrected the symbol of LI and HI in Figure 1A (Page 4).
Point 5: I would suggest the author to give some idea what kind of surveillance would be more feasible according to the present study.
Response 5: Thank you for this feedback. We have revised the concluding section to reflect your advice, and the revised conclusion clarifies the feasibility of surveillance that can be suggested based on the results of this study (Page 11, line 3).
Reviewer 2 Report
Major revisions:
1- There are many differences in the two groups that can affect the DFS and OS. Cox regression for DFS and OS including all factors would be better.
2- Why the authors included only patients who did perioperative CT?
3-please better explain the inclusion criteria (did you use an institutional database?)
4-the conclusions are too strong for the methods (retrospective and so on).
Minor:
Some English check is needed
Author Response
Response to Reviewer 2 Comments
We thank you for the time and effort dedicated in thoroughly reviewing our manuscript and for the detailed and specific comments that have helped us considerably improve the presentation of our research. We have carefully revised our manuscript as suggested.
Point 1: There are many differences in the two groups that can affect the DFS and OS. Cox regression for DFS and OS including all factors would be better.
Response 1: Thank you for this crucial advice. We sincerely apologize for the mistake. The multivariate analysis for DFS and OS presented in Table 2 was performed using the Cox regression method, which included all factors that could be considered as time-related variables. We have corrected this error in the Statistical analyses subsection of the Materials and methods (Page 10, line 32).
Point 2: Why the authors included only patients who did perioperative CT?
Response 2: Thank you for your query. This study has a retrospective design and was conducted using institutional data; moreover, all patients in our study center underwent surveillance based on serum CEA levels, abdominopelvic CT, chest CT, and colonoscopy at regular intervals. Therefore, we determined the surveillance intensity on the basis of the number of CT scans, which was a common imaging test that all patients underwent.
Point 3: Please better explain the inclusion criteria (did you use an institutional database?)
Response 3: Thank you for indicating this point. Accordingly, we clarified that we included all pathologic stage I to III CRC patients who underwent radical resection for primary CRC. The inclusion criteria have been specified in the Patients and clinical variables subsection in the Materials and methods section (Page 9, line 28).
Point 4: The conclusions are too strong for the methods (retrospective and so on).
Response 4: Thank you for this feedback. We have accordingly revised the conclusion to accurately reflect the results of the study (Page 11, line 3).
Point 5: Some English check is needed.
Response 5: We corrected English and submitted the revised paper with certificate.
Reviewer 3 Report
The manuscript entitled "Optimal postoperative surveillance strategies for colorectal cancer: a retrospective observational study" highlgihted that Frequent postoperative imaging alone in CRC patients does not improve OS or RFS. A higher imaging frequency in patients at high risk of recurrence could be expected to lead to a slight increase in PRS but does not improve OS. Therefore, in high-risk patients, it is not helpful to increase the number of CT scans, which is currently recommended for surveillance. There is a need to explore other imaging modalities or innovative ap-proaches such as liquid biopsy and evaluate the effect of these tools on oncologic out-comes and facilitate the early detection of recurrence.
The manuscript is really well written and of interest for the audience.
Minor comments:
- The manuscript may benefit from a slight language revision.
- The Authors should provide the expand forms for all acronyms through the text when they first appear.
Author Response
Thank you for your comments. We provided the expand forms for missed acronyms in the text.
Reviewer 4 Report
The manuscript can be accepted.
Author Response
Thank you for your decision.
Round 2
Reviewer 1 Report
Thank you to the author for the improvement of the manuscript. Most of the question and suggestion seem to be improved by the author. However, I cannot agree to the author that with the word "phenomenon". I feel that this discussion is limited in this study and we cannot reach to the conclusion. To overcome this severe concern, I think we need more subgroup analysis (for example, Stage I, II, III), but the problem would be the patient number and the recurrence in each subgroup. The patient number might be relatively small to conclude and it might be difficult to understand this phenomenon.
Author Response
Thank you for your very insightful comments. We completely agree with your concern and the word “phenomenon” was inappropriate. This was intended to express the meaning of our current medical condition. But it was a hasty conclusion, as you said, to bring our results to a conclusion. We performed subgroup analysis according to your suggestion, and the results are shown in the following table. We added this result as supplementary table in the Result section, and the conclusion was revised according to this result.
Table. Subgroup analysis according to the stage.
|
Variables |
Surveillance intensity |
P-value |
|
|
Lower intensity |
Higher intensity |
||
|
Stage I (N=1131) |
N=953 |
N=178 |
|
|
Overall recurrence, No (%) |
24 (2.5) |
18 (10.1) |
<.001 |
|
Overall survival, months, mean (SD) |
59.07 (18.33) |
60.15 (18.59) |
.474 |
|
Recurrence free survival, months, mean (SD) |
57.38 (19.32) |
55.34 (21.37) |
.203 |
|
Post recurrence survival, months, mean (SD) |
31.83 (27.79) |
40.44 (23.72) |
.286 |
|
Stage II (N=1277) |
N=818 |
N=459 |
|
|
Overall recurrence, No (%) |
56 (6.8) |
69 (15.0) |
<.001 |
|
Overall survival, months, mean (SD) |
55.73 (21.57) |
58.18 (21.73) |
.054 |
|
Recurrence free survival, months, mean (SD) |
52.06 (23.40) |
50.57 (25.04) |
.288 |
|
Post recurrence survival, months, mean (SD) |
32.73 (22.96) |
41.19 (21.54) |
.037 |
|
Stage III (N=1187) |
N=535 |
N=652 |
|
|
Overall recurrence, No (%) |
95 (17.8) |
172 (26.4) |
<.001 |
|
Overall survival, months, mean (SD) |
53.86 (24.77) |
26.79 (21.71) |
.030 |
|
Recurrence free survival, months, mean (SD) |
47.69 (27.38) |
47.83 (25.17) |
.931 |
|
Post recurrence survival, months, mean (SD) |
27.16 (22.44) |
31.78 (22.80) |
.110 |
No, number; SD, standard deviation
Reviewer 2 Report
I appreciate the corrections the authors made.
I find the work clear and methodologically well done.
Author Response
Thank you for your decision.